# Plasma cytokine profiles in very preterm infants with late-onset sepsis

Julie Hibbert[1,2], Tobias Strunk[1,2,3], Karen Simmer[1,2,3], Peter Richmond[1,2], David Burgner[4,5], Andrew Currie[1,6]*

1 Centre for Neonatal Research and Education and Division of Paediatrics, Medical School, University of Western Australia, Perth, Western Australia, Australia, 2 Telethon Kids Institute, Perth, Western Australia, Australia, 3 Neonatal Directorate, King Edward Memorial Hospital, Perth, Western Australia, Australia, 4 Murdoch Children's Research Institute, Royal Children's Hospital, Melbourne, Victoria, Australia, 5 Department of Paediatrics, University of Melbourne, Melbourne, Victoria, Australia, 6 Medical, Molecular and Forensic Sciences, Murdoch University, Perth, Western Australia, Australia

* A.currie@murdoch.edu.au

**Data Availability Statement:** All relevant data are within the paper and Supporting Information files.

**Funding:** This study was supported by the National Health and Medical Research Council (NHMRC) of Australia (572548, www.nhmrc.gov.au). TS is

## Abstract

### Introduction

Deficiencies in innate immune responses may contribute to the increased susceptibility to infection in preterm infants. *In vivo* cytokine profiles in response to sepsis in very preterm infants are not fully understood.

### Aims

To characterise plasma pro- and anti-inflammatory cytokine concentrations and pre-defined ratios in very preterm infants with late-onset sepsis (LOS).

### Methods

In this observational study, peripheral blood samples were collected at the time of evaluation for suspected LOS from 31 preterm infants (<30 weeks gestational age). Plasma cytokine concentrations were determined by 12-plex immunoassay.

### Results

IL-10, IFN-γ, IL-12p70, IP-10, IL-6 and CCL2 were elevated in the majority infants with LOS (n = 12) compared to those without LOS (n = 19). There was no difference in TNF-α, IL-1β, IL-17AF, IL-8 and IL-15 concentrations between groups. IL-10/TNF-α ratios were increased, while CCL2/IL-10 and IL-12p70/IL-10 ratios were decreased in infants with LOS compared to those without.

### Conclusion

Very preterm infants have a marked innate inflammatory response at the time of LOS. The increase in IL-10/TNF-α ratio may indicate early immune hypo-responsiveness. Longitudinal studies with a larger number of participants are required to understand immune responses and clinical outcomes following LOS in preterm infants.

supported by a Raine Medical Research Foundation Clinician Research Fellowship (rainefoundation.org.au). DB is supported by NHMRC Research Fellowship (www.nhmrc.gov. au). JH is supported by a University Postgraduate Award (www.education.gov.au). The funders had no role in study design, data collection and analysis, decision to publish, or preparation of the manuscript.

**Competing interests:** The authors have declared that no competing interests exist.

## Introduction

Globally, approximately 15 million infants are born preterm (<37 weeks gestational age, GA) each year, of which ~15% are born less than 32 weeks GA. [1] Preterm infants are susceptible to invasive bacterial infections and the risk is associated with lower GA and birthweight. [2] Despite advances in neonatal care, late-onset sepsis (LOS; occurring >72 hours after birth) is a frequent complication in preterm infants, affecting up to 40% of infants born <28 weeks GA. [2] Infection-related inflammation is associated with increased mortality and morbidity, particularly long-term adverse neurodevelopmental outcomes. [3]

The mechanisms underpinning the distinct neonatal immune responses to bacterial infections are incompletely understood. Pro- and anti-inflammatory cytokines are critical key initiators and regulators of inflammation and host response to bacterial infection. [4] In adults, both pro- and anti-inflammatory cytokines are simultaneously produced during the early stages of sepsis and correlate with disease progression and mortality. [5] Elevated ratios of anti- and pro-inflammatory cytokines (e.g. IL-10/TNF-$\alpha$ and IL-6/IL-10) are associated with multiple organ failure and are proposed markers of sepsis-induced immunosuppression in adult patients following sepsis. [6, 7] Recovery from sepsis may depend on the balance of pro- and anti-inflammatory immune responses to achieve homeostasis. [8]

It is unclear whether the immediate response of preterm infants with LOS is hyper- or hypo-inflammatory. [9] Cytokine concentrations at the time of neonatal sepsis are inconsistent, partly as studies often include both LOS and early-onset sepsis (EOS; occurring <72 hours after birth), [10–12] and a wide range of GA, [11, 13–18] making interpretation of cytokine responses challenging. We therefore aimed to characterise key cytokines [5] in plasma at the time of evaluation for suspected LOS in a homogeneous cohort of very preterm infants (born <30 weeks GA) enrolled in an observational cohort study of innate immune ontogeny. We also investigated whether infants with LOS have increased pro- and anti-inflammatory cytokine responses and elevated pre-defined cytokine ratios, as observed in adults, compared to neonates without LOS.

## Materials and methods

### Study participants

This study was approved by the Women and Newborn Health Services Human Research Ethics Committee, Perth, Australia (1627/EW). Written, informed consent was obtained by the Principal Investigator or delegate from the parent(s)/legal guardian prior to study participation. Infants born less than 30 weeks GA without birth defects or genetic abnormalities who were admitted to the Neonatal Intensive Care Unit (NICU) at King Edward Memorial Hospital, Perth, Australia were eligible to participate in this prospective observational cohort study of innate immune system ontogeny, with recruitment from July 2009 to October 2011. Samples taken at the time of evaluation for suspected LOS were collected opportunistically during normal working hours from enrolled infants. Blood culture was performed on infants if they had clinical signs suggestive of LOS, including increased ventilation or oxygen requirements, lethargy, decreased perfusion, temperature instability, vomiting, feed intolerance, increased apnoea and/or bradycardia. For ethical reasons only one blood sample was collected at the time of sepsis evaluation, when the clinical decision was made to investigate sepsis and commence antibiotic therapy. Infants who had episodes of EOS (n = 2) or necrotising enterocolitis (NEC) prior to the episode of suspected LOS (n = 1) were excluded. Placental histology were analyzed using an adaptation of a widely accepted semi-quantitative scoring system, as previously described. [19] Haematological parameters were recorded close to the time of blood culture sampling (mean ± SD, 124 ± 177 min).

### Definition of late-onset sepsis

'Confirmed LOS' was retrospectively defined as a positive blood culture with a single organism and a C-reactive protein (CRP) of >15mg/L within 72hrs of blood culture sampling and anti-biotic therapy for ≥ 5 days, as per the local NICU guidelines during the study period. Episodes without evidence for LOS, referred to as 'no LOS', were defined as a negative blood culture. Four infants were considered to have contaminating blood cultures (based on the presence of Gram-positive commensal bacteria, lack of an inflammatory response with two or more CRP measurements <15mg/L and absence of clinical features of sepsis) were excluded from analysis.

### Blood sampling and processing for cytokine analyses

Approximately 0.5ml of peripheral blood was collected by venepuncture into Lithium-heparin tubes (BD Biosciences, North Ryde, Australia) at the time of blood culture (68% of samples) or close to the time of blood culture sampling (mean ± SD, 1.2 ± 2.6 hr). Sample processing and analysis was performed at the Children's Clinical Research Facility in Perth, Australia. Samples were centrifuged at 6,000 x $g$ for 2 minutes, and plasma extracted and stored at -80˚C until cytokine measurement.

### Multiplex immunoassay

Plasma cytokine concentrations were quantified using a custom 12-plex magnetic bead-based immunoassay kit (ProcartaPlex Biosystems eBioscience, San Diego, California) following the manufacturer's instructions. The kit provided a mixture of magnetic beads conjugated with primary antibodies against IL-1β, IL-6, IL-12p70, IL-15, IL-17AF, IL-8, IL-13, IL-10, TNF-α, interferon (IFN)-γ, inducible protein (IP)-10 and chemokine (C-C motif) ligand-2 (CCL2). Fluorescence for each cytokine bead region was acquired electronically in real time on the Bio-Plex 200 System (Bio-Rad, Gladesville, Australia) and analysed using BioPlex Manager 5.0 Software. Cytokine concentrations, in pg/mL, were generated from a seven-point, five-param-eter (four-parameter for IL-13) logistic standard curve. All samples analysed were run as a sin-gle sample on one plate. Concentrations above the highest standard were repeated at a dilution that fell within the standard curve. Concentrations below the lowest standard were assigned a value of half the lowest standard for statistical analysis.

### Statistical analysis

Continuous data were summarised with median and interquartile ranges (IQR, 25th-75th) and categorical data with frequency distributions. All cytokine data were tested for normality using the Shapiro-Wilk normality test and presented using nonparametric data summaries, medians and IQR (25th-75th percentiles). Comparisons between the 'confirmed LOS' and 'No LOS' groups were by the Mann-Whitney test for continuous data and Chi-square tests for categori-cal data. All statistical analysis was performed using Prism 8 software (Graphpad Software, San Diego, California). $P$ values <0.05 were considered significant.

## Results

### Study subject characteristics

There were no differences in the basic demographic and clinical details between the infants with confirmed LOS (n = 12) and no LOS (n = 19), details are shown in Table 1. One infant from the confirmed LOS group (sampled at 16 days postpartum) died aged 44 days of respira-tory failure, and two infants in the no LOS group (both sampled at day 7 postpartum) died

**Table 1. Basic demographic and clinical parameters of study cohort[a].**

|  | Confirmed LOS (n = 12) | No LOS (n = 19) | *P* value |
|---|---|---|---|
| Gestational age (weeks) | 26.7 (24.5–28.4) | 25.6 (24.6–26.6) | 0.383 |
| Birthweight (grams) | 835 (669–1101) | 705 (555–895) | 0.194 |
| Male | 4 (33.3) | 12 (63.2) | 0.149 |
| Caesarean section | 5 (41.7) | 11 (57.9) | 0.473 |
| Membranes rupture >24h before delivery | 4 (33.3) | 9 (47.4) | 0.484 |
| Antenatal steroids[b] | 11/12 (91.7) | 18/18 (100) | 0.400 |
| Histological chorioamnionitis[b] | 5/11 (45.5) | 9/16 (56.3) | 0.704 |
| Mechanical ventilation | 12 (100) | 19 (100) | >0.999 |
| Duration (hours) | 241 (26.3–842) | 394 (115–1091) | 0.417 |
| CPAP | 12 (100) | 18 (94.7) | >0.999 |
| Duration (hours) | 1003 (771–1284) | 1057 (563–1298) | 0.723 |
| IVH (grade III/IV) | 0 (0) | 2 (10.5) | 0.510 |
| ROP (stage III/IV) | 0 (0) | 1 (5.3) | >0.999 |
| Length of NICU stay (days) | 83 (60–109) | 109 (94–149) | 0.150 |

[a]Data are expressed as median (IQR) or n (%), as appropriate.

[b]From available reports. CPAP, continuous positive airway pressure; IVH, intraventricular haemorrhage; ROP, retinopathy of prematurity

aged 19 and 56 days of NEC and respiratory failure, respectively. No other infants in the no LOS group developed infections or NEC prior to discharge from the NICU.

The clinical parameters at the time of sepsis evaluation are shown in Table 2. For the 12 episodes of confirmed LOS, coagulase-negative staphylococci (CoNS) were the most common infectious agents isolated (10 cases, 83%), consistent with local epidemiology. [20] All infants recovered from the LOS episode.

**Table 2. Clinical parameters at the time of sepsis evaluation [a].**

|  | Confirmed LOS (n = 12) | No LOS (n = 19) | *P* value |
|---|---|---|---|
| Age (days) | 12 (9–14) | 10 (7–15) | 0.404 |
| Gram-positive organisms |  |  |  |
| CoNS | 10 (83.4) | - | - |
| *Bacillus sphaericus* | 1 (8.3) | - | - |
| *Enterococcus faecalis* | 1 (8.3) | - | - |
| CRP (mg/L) |  |  |  |
| Highest within 72hr blood culture | 59 (26–91) | 17 (5–30) | 0.0008 |
| At blood culture sampling[b] | 20 (7–36) | 12 (5–24) | 0.106 |
| White blood cell count (x10^9/L)[c] | 14.5 (9.4–18.1) | 24.8 (15.0–31.5) | 0.095 |
| Neutrophil count (x10^9/L)[c] | 11.1 (5.0–15.4) | 16.7 (9.6–20.0) | 0.244 |
| Platelet count (x10^9/L)[c] | 137 (102–219) | 215 (118–332) | 0.253 |
| Mechanical ventilation commenced | 4 (33.3) | 5 (26.3) | 0.704 |
| Inotrope use | 0 (0) | 0 (0) | >0.999 |

[a]Data are expressed as median (IQR) or n (%), as appropriate.

[b]At or before (mean ± SD, 2.5 ± 2.5 hr) blood culture sampling.

[c]Data available for n = 11 confirmed LOS and n = 18 no LOS. CoNS, coagulase-negative staphylococci; CRP, C-reactive protein.

**Table 3.** Circulating cytokine concentrations at the time of late-onset sepsis evaluation[a].

| | Multiplex pg/mL range | | Confirmed LOS (n = 12 episodes) | | | No LOS (n = 19 episodes) | | | P value |
|---|---|---|---|---|---|---|---|---|---|
| | Minimum | Maximum | Median | IQR | Range | Median | IQR | Range | |
| IL-6 | 9.2 | 37,800 | 231 | 62.8–922 | 4.6–4309 | 4.6 | 4.6–149 | 4.6–2881 | 0.017 |
| IFN-γ | 12.4 | 50,700 | 347 | 65.7–712 | 6.2–1512 | 6.2 | 6.2–6.2 | 6.2–949 | 0.001 |
| IL-12p70 | 6.9 | 28,100 | 32.8 | 3.4–72.7 | 3.4–85.7 | 3.4 | 3.4–24.4 | 3.4–39.3 | 0.049 |
| TNF-α | 7.2 | 29,500 | 9.1 | 6.1–32.2 | 3.6–49 | 6.1 | 3.6–14.0 | 3.6–23.2 | 0.168 |
| IL-15 | 3.1 | 12,500 | 43.8 | 1.5–383 | 1.5–576 | 218 | 1.5–276 | 1.5–670 | 0.676 |
| IL-1β[b] | 2.0 | 8,250 | 1.0 | 1.0–9.7 | 1.0–164 | 1.0 | 1.0–3.8 | 1.0–33.1 | 0.861 |
| IL-17AF[b] | 6.1 | 25,000 | 3.1 | 3.1–332 | 3.1–960 | 3.1 | 3.1–3.1 | 3.1–347 | 0.172 |
| IL-8 | 2.2 | 8,900 | 234 | 137–1103 | 51.1–5138 | 174 | 84.2–707 | 33.7–6628 | 0.164 |
| IP-10 | 2.2 | 8,800 | 13079 | 8348–16026 | 2661–18013 | 1066 | 602–1498 | 57.6–16033 | <0.0001 |
| CCL2 | 4.8 | 19,500 | 3407 | 2428–7627 | 1780–15105 | 1350 | 473–2588 | 184–7701 | 0.002 |
| IL-10 | 2.3 | 9,250 | 194 | 46.8–255 | 22.9–487 | 6.5 | 1.1–22.9 | 1.1–529 | <0.0001 |
| IL-13[b] | 3.3 | 13,400 | 1.6 | 1.6–3.8 | 1.6–16.5 | 1.6 | 1.6–1.6 | 1.6–1.6 | 0.049 |

[a]Data are expressed as median, IQR (25th-75th) and range in pg/mL.

[b]Fewer than 25% of data points in both groups are above the lowest detectable standard. LOS, late-onset sepsis; IL, interleukin; IFN, interferon; IP, inducible protein, CCL2, chemokine (C-C motif) ligand-2; TNF, tumour necrosis factor; ns, not statistically significant. Level of significance measured by Mann-Whitney test.

## Circulating cytokine concentrations at the time of late-onset sepsis evaluation

Overall, infants with confirmed LOS had markedly higher concentrations of IL-10, CCL2, IFN-γ, IL-12p70, IL-13, IL-6 and IP-10 compared to no LOS, whereas concentrations of TNF-α, IL-15, IL-1β, IL-17AF and IL-8 were similar in the two groups (Table 3). For IL-1β, IL-17AF and IL-13, fewer than 25%, 15% and 10%, respectively, of concentrations were above the lowest limit of detection, and results should be interpreted accordingly.

## Analysis of anti- and pro-inflammatory ratios in infants with suspected late-onset sepsis

We next evaluated whether the ratios of anti- and pro-inflammatory cytokine concentrations, namely IL-10/TNF-α, IL-6/IL-10 and IP-10/IL-10, differed between groups, as previously reported in adults [6] and infants. [12–14] Infants with confirmed LOS had a markedly elevated IL-10/TNF-α and IP-10/IL-10 ratio compared to infants with no LOS (median (IQR) 10 (5–38) vs 0.9 (0.3–3), and 72 (38–144) vs 188 (82–533), respectively; Fig 1A and 1B), yet IL-6/IL-10 was similar across groups (Fig 1C).

As concentrations of the pro-inflammatory cytokine IFN-γ, IL-12p70 and chemokine CCL2 were associated with confirmed LOS, we evaluated the ratios of CCL2 to IL-10, IFN-γ to IL-10 and IL-12p70 to IL-10 to determine if an anti- or pro-inflammatory response predominated. A low ratio of CCL2/IL-10 and IL-12p70/IL-10 were strongly associated with confirmed LOS compared to the no LOS group (median (IQR) 38 (13–78) vs 286 (52–563) and 0.2 (0.03–0.9) vs 1.9 (0.5–3.0), respectively. Fig 1D and 1E), but there was no difference in the IFN-γ/IL-10 ratio between the patient groups (Fig 1F).

## Discussion

In this study, we report that very preterm infants with microbially confirmed LOS had elevated concentrations of both pro-inflammatory cytokines, IFN-γ, IP-10, IL-12p70, IL-6, CCL2 and

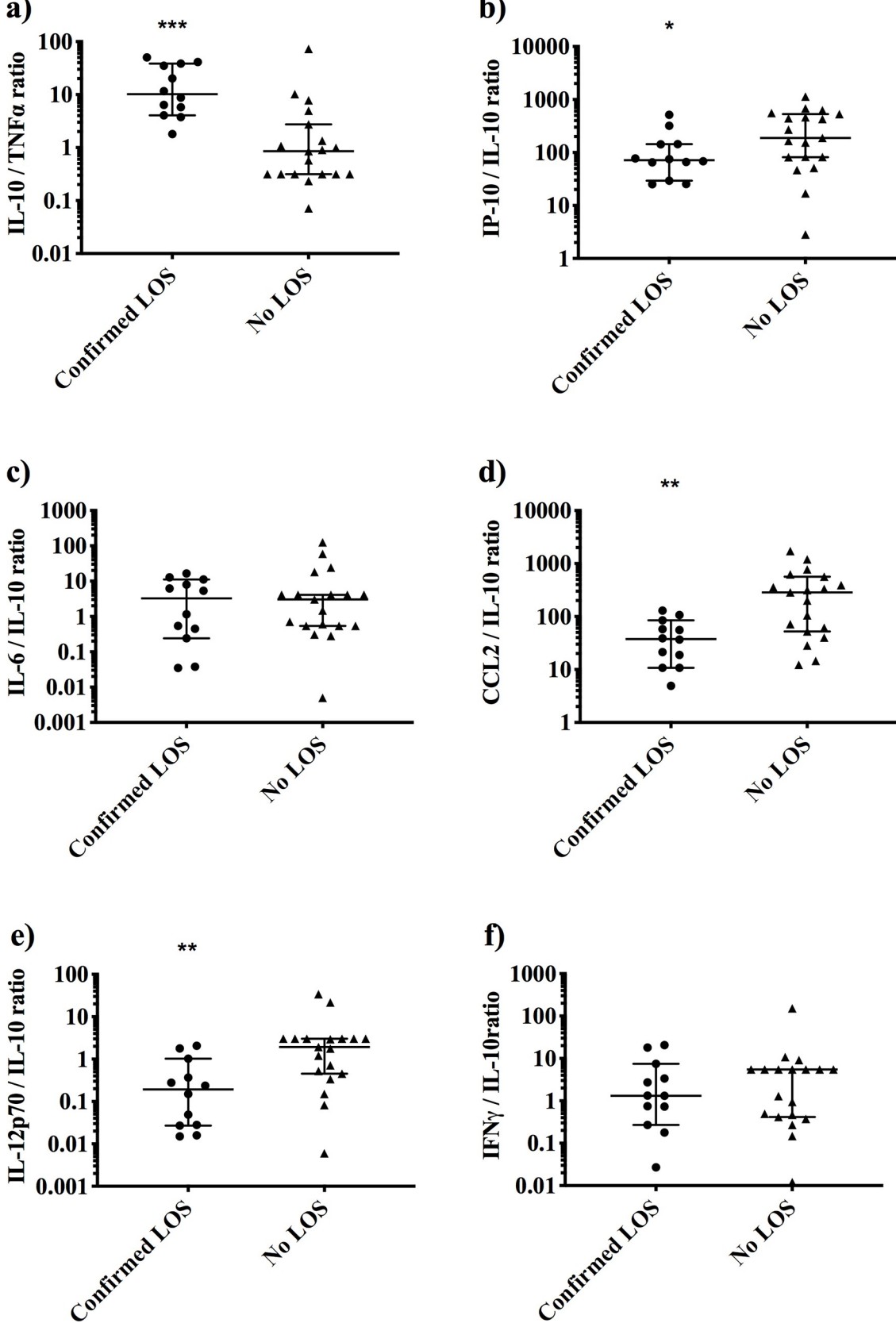

**Fig 1. Cytokine ratio at the time of late-onset sepsis evaluation.** Comparison of **a**) IL-10/TNF-α, **b**) IP-10/IL-10, **c**) IL-6/IL-10, **d**) CCL2/IL-10, **e**) IL-12p70/IL-10 and **f**) IFN-γ/IL-10 from infants with confirmed LOS (n = 12 episodes, closed circles) and no LOS (n = 19 episodes, closed triangles). Cytokines were measured by bead-based multiplex assay. All data shown on a log scale with median with 95% confidence intervals. Symbols depict level of significance between confirmed LOS and no LOS groups (*P = 0.048; **P = 0.003; ***P = 0.0002) by Mann-Whitney test. LOS, late-onset sepsis; IL, interleukin; IFN, interferon; IP, inducible protein; CCL2, chemokine (C-C motif) ligand-2.

the anti-inflammatory cytokine IL-10, compared to those without LOS. TNF-α levels were relatively reduced in infants with LOS. Furthermore, we observed an elevated IL-10/TNF-α ratio and decreased CCL2/IL-10 and IL-12p70 ratios in infants with confirmed LOS, compared to the no LOS group.

## Elevated circulating cytokine concentrations are associated with late-onset sepsis

Elevated plasma concentrations of IL-10, IFN-γ, IP-10, IL-12p70, IL-6 and CCL2, were strongly associated with confirmed LOS compared to no LOS; consistent with other preterm infant sepsis studies. [10, 13, 14, 16, 18]. Concentrations of TNF-α, IL-1β, IL-17AF and IL-8 were similar between the two groups. Published concentrations of TNF-α, IL-1β, IL-17AF and IL-8 in infants with sepsis are inconsistent, with some studies reporting results similar to ours, [10–12, 15, 17] and others reporting higher cytokine concentrations in septic infants. [14–17] The reasons underlying these discrepant results are unclear. Several factors may affect cytokine concentrations, including GA, [10] EOS vs LOS (which generally have different aetiologies), [12] sepsis definition (culture proven vs clinical sepsis) [15] and inclusion of Gram-negative, fungal and NEC-related infections (which may elicit different cytokine responses). [13] To our knowledge, this is the first study to measure IL-15 in neonates with and without sepsis. IL-15 has anti-apoptotic properties that may prevent sepsis-induced apoptosis of immune cells in septic patients, [21]. Similar to adult sepsis studies, [22] we did not observe a difference in IL-15 between preterm infants with and without sepsis.

## Cytokine ratios as potential indicators of neonatal late-onset sepsis

The elevated ratio of IL-10/TNF-α in infants with confirmed LOS may indicate early immune hypo-responsiveness. Elevated ratio of IL-10/TNF-α has been observed in adult sepsis data and suggested to indicate sepsis-induced immunosuppression, [6]; it may therefore indicate relative immunosuppression in preterm infants with sepsis. The balance between these cytokines may be important in sepsis pathogenesis and susceptibility to infections in preterm infants, as described in other children and adults with sepsis. [6, 23] Other studies report markedly lower levels of LPS-, R848-, and LPS/ATP-induced TNF-α/IL-10 ratios in neonatal cord blood compared to adult peripheral blood, and an increased LPS-stimulated IL-10/TNF-α ratio in cord blood from infants born very preterm compared to moderate and late preterm. [24, 25] Further, peripheral blood from infants with LOS had a markedly higher IL-10/TNF-α ratio following stimulation with live *Staphylococcus epidermidis* compared to uninfected infants, indicating infected infants have an impaired immune response to bacteria. [26] This suggests that a marked anti-inflammatory response may be characteristic of infants and particularly those who are very preterm. Our results of elevated IL-10/TNF-α and decreased IP-10/IL-10 ratios are similar to other studies looking at septic neonates with disseminated intravascular coagulation (DIC). [13, 14]

We did not observe a difference in the IL-6/IL-10 ratio between groups. Previous studies report an elevated IL-6/IL-10 ratio in septic infants, [12] and specifically in those with

disseminated intravascular coagulation, which is often associated with fungal and Gram-negative bacterial infections that are known to elicit different cytokine responses. [13, 14] The infants with LOS in our study had Gram-positive bacterial LOS and did not have DIC, which may explain why an elevated ratio was not observed in our cohort. Further studies with well-defined sepsis definitions are needed to resolve these discrepant findings.

The elevated CCL2 and IL-12p70 observed in LOS infants is consistent with data on neonatal [11, 14, 16] and adult sepsis. [5] To our knowledge, this is the first study to report lower ratios of CCL2/IL-10 and IL-12p70/IL-10 in infants with LOS compared to infants without LOS. The role of CCL2, a member of the chemokine CC subfamily with potent monocyte chemotactic activity, in sepsis is unclear. In a murine model of caecal ligature and puncture-induced infection, higher levels of CCL2 modulates peritoneal bacterial clearance, suggesting CCL2 is important in antimicrobial defence. [27] In contrast, adult studies report that sepsis mortality is associated with increased CCL2, [5] potentially due to chemotaxis of innate leukocytes and marked production of pro-inflammatory cytokines that contribute to organ failure, septic shock and death. [28] Elevated levels of IL-12p70, an IFN-γ inducing pro-inflammatory cytokine, have been reported in other preterm infant sepsis studies. [18] Further studies investigating the role of CCL2 and IL-12p70 in LOS in preterm infants are warranted.

## Study strengths and limitations

This is the first study to show a skewed anti-inflammatory response in very preterm infants with only Gram-positive LOS. Our study focused on a well-defined population of very preterm infants with LOS, which may reduce some of the inter-individual variability seen in other neonatal studies. We acknowledge some limitations, including modest sample size, which precluded investigation of rarer Gram-negative and fungal associated LOS and infrequent outcomes, such as septic shock and death. We were unable to collect serial samples over the course of the episode and therefore lack important kinetic data. Lastly, it was not considered ethical to take blood samples from very preterm infants without clinical indication, so the no LOS group had clinical signs that led to investigations for sepsis at the time of sample collection. This group, therefore, may not be representative of responses in otherwise healthy very preterm infants.

## Conclusions

Very preterm infants with LOS have elevated concentrations of both pro- and anti-inflammatory cytokines with a prominent shift to an anti-inflammatory immune response. These results suggest that neonatal LOS maybe associated with a hypo-responsiveness, similar to sepsis-induced immunosuppression observed in adults. Consistently reported patterns of cytokines production associated with sepsis may lead new diagnostic methods, prognostic tools and highlight possible therapeutic targets. Future studies should further explore the kinetic profile of circulating cytokines in preterm infants with sepsis by serial sampling over the course of the septic episode and ideally would be powered to evaluate associations between cytokine profile and disease severity and outcomes.

## Acknowledgments

Authors would like to acknowledge the assistance of Gail Abernethy, Annie Chang, Chooi Heen Kok and the nursing staff at King Edward Memorial Hospital NICU for recruitment and sample collection. We would like to thank Professor Dorota Doherty and Ms Liz Nathan (both of University of Western Australia) for statistical advice. Lastly, we would like to thank all the study participants and their families.

## Author Contributions

**Conceptualization:** Julie Hibbert, Tobias Strunk, Karen Simmer, Peter Richmond, David Burgner.

**Formal analysis:** Julie Hibbert.

**Funding acquisition:** Tobias Strunk, Karen Simmer, Peter Richmond, David Burgner, Andrew Currie.

**Investigation:** David Burgner, Andrew Currie.

**Methodology:** Julie Hibbert, Tobias Strunk, Andrew Currie.

**Project administration:** Julie Hibbert.

**Supervision:** Tobias Strunk, Karen Simmer, David Burgner, Andrew Currie.

**Writing – original draft:** Julie Hibbert.

**Writing – review & editing:** Tobias Strunk, Karen Simmer, Peter Richmond, David Burgner, Andrew Currie.

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
