## [Decision Letter · Decision Letter 0]

31 Dec 2019

PONE-D-19-33625

Plasma cytokine profiles in very preterm infants with late-onset sepsis

PLOS ONE

Dear Ms Hibbert,

Thank you for submitting your manuscript to PLOS ONE. After careful consideration, we feel that it has merit but does not fully meet PLOS ONE’s publication criteria as it currently stands. Therefore, we invite you to submit a revised version of the manuscript that addresses the points raised during the review process.

We would appreciate receiving your revised manuscript by Feb 14 2020 11:59PM. To enhance the reproducibility of your results, we recommend that if applicable you deposit your laboratory protocols in protocols.io, where a protocol can be assigned its own identifier (DOI) such that it can be cited independently in the future. For instructions see: http://journals.plos.org/plosone/s/submission-guidelines#loc-laboratory-protocols

We look forward to receiving your revised manuscript.

Kind regards,

Philip Alexander Efron, MD, FACS, FCCM

Academic Editor

PLOS ONE

Journal Requirements:

2. In your Methods section, please provide additional information about the participant recruitment method and the demographic details of your participants. Please ensure you have provided sufficient details to replicate the analyses such as: a) the recruitment date range (month and year), b) a description of any inclusion/exclusion criteria that were applied to participant recruitment, c) a description of how participants were recruited, and d) descriptions of where participants were recruited and where the research took place.

Reviewers' comments:

Reviewer's Responses to Questions

**Comments to the Author**

1. Is the manuscript technically sound, and do the data support the conclusions?

Reviewer #1: Yes

Reviewer #2: Partly

Reviewer #3: Partly

2. Has the statistical analysis been performed appropriately and rigorously? 

Reviewer #1: Yes

Reviewer #2: Yes

Reviewer #3: Yes

3. Have the authors made all data underlying the findings in their manuscript fully available?

Reviewer #1: Yes

Reviewer #2: Yes

Reviewer #3: Yes

4. Is the manuscript presented in an intelligible fashion and written in standard English?

Reviewer #1: Yes

Reviewer #2: Yes

Reviewer #3: Yes

5. Review Comments to the Author

Reviewer #1: Title: Plasma cytokine profiles in very preterm infants with late-onset sepsis

Journal: PLOS ONE

Summary: Thank you for the opportunity to review this manuscript by Hibbert et al. The authors present an interesting study in which blood samples were obtained from 31 preterm neonates at the time of evaluation for late-onset sepsis. Plasma cytokines concentrations were determined, and inferences were made about the status of neonatal immune responses. The authors should be applauded for performing the study given the difficulties in blood collection in this vulnerable population, and the need for improved understanding of the neonatal immune response. Overall the study is very well presented, but could benefit from some clarification and potentially a multivariate logistic regression.

Major Points:

1. Why were patients with necrotizing enterocolitis excluded from the study? This is an important source of gram-negative, late-onset sepsis. Is this disease process considered fundamentally different from the LOS cases studied?

2. In Methods, Blood sampling: blood was collected close to the time of blood culture sampling. How was blood collected and how much was drawn? Was the blood draw performed separately from the blood culture collection? Was it a capillary heelstick, venipuncture, or drawn from an existing line?

3. Could the composite cytokine score ratio of anti- to pro-inflammatory cytokines be biased by the fact that there are only two anti-inflammatory cytokines (IL-13 and IL-10) included compared to many more (10) pro-inflammatory cytokines in the ratio? Since sums of the normalized scores are used for the ratio, it seems that the ratio will always be small given the large sums in the denominator for 10 different inflammatory cytokines.

4. Did the authors consider performing multivariate logistic regression to predict sepsis vs. non-sepsis? This is a well-performed prospective cohort study with fairly well-balanced sepsis vs. non-sepsis groups. It would be interesting to evaluate which clinical and cytokine factors emerge as the most significant predictors of sepsis, as this could help determine which cytokines should be measured to assist with the diagnosis of neaonatal sepsis. If possible, I would suggest having sepsis vs. non-sepsis be the outcome of the regression, and include multiple explanatory varaibles (such as IL-10, IFN-gamma, gestational age at birth, day of life, CRP, platelet count). This would aid in creating a predictive model with an area-under-the-curve to predict late-onset sepsis based on parameters that can be obtained from POC tests/clinical information. This would also lend support to statements in the conclusion about the association of cytokine levels with sepsis vs. non-sepsis.

Minor Points:

1. Table 1 has a line describing “mortality during episode”. What does this mean, i.e. how is the episode defined? I would consider deleting this line given that there is already a descriptive statement in the results stating that only one infant died 12 days after the initial sample collection.

2. Table 2: Please include the calculated p-value rather than “ns” as this offers a more complete picture of the analysis. For example, median IL-15 levels were 43.8 in the sepsis group and 218 in the non-sepsis group. A p-value rather than “ns” here would help with interpretation of that result.

Recommendation: Major Revision

Reviewer #2: The manuscript, “Plasma cytokine profiles in very preterm infants with late-onset sepsis” evaluates various cytokine profiles in the plasma of very preterm infants. They found that there are marked innate inflammatory responses, with a relatively high anti-inflammatory to pro-inflammatory ratio. While the understanding of immune responses in preterm infants is important, a number of issues need to be addressed prior to publication:

1. Can the authors state very clear inclusion and exclusion criteria in the methods section? There is a mention in the limitations that the “no LOS” group were patients that had concerns for sepsis, but ultimately did not have a positive blood culture. Without further description, this is not necessarily an adequate comparison group. Was there any possibility to collecting blood from healthy pre-term infants or did the authors feel this was not clinically ethical? In addition, the authors state that one infant from the “no LOS” group had NEC. Did any of the other “no LOS” patients have an infection? This would be extremely important to know, as not all septic patients will have positive blood cultures and/or bacteremia may be transient enough to miss.

2. Can the authors state why they chose a single early time point and what was the goal for sample collection? In the methods section, they state that, “hematological parameters were recorded close to the time of blood culture sampling” of about 3h (124 ±177min). Was the time of blood culture sampling presumably time zero of diagnosis of LOS?

3. Although the authors state that 83% had gram positive cultures, can the remainder of the organism types be listed (number and %). Were there any causes for the bacteremia (ie what was the infectious source? Was this primary bacteremia or from other sources)?

4. While the LOS group had significantly higher levels of both pro- and anti-inflammatory cytokines, this does not necessarily predict immune cell function and therefore one cannot conclude that the patients are “immunosuppressed”. Do the authors have any insight into actual immune cell function?

5. In the “multiplex immunoassay” section in the Methods, the authors state that cytokine concentrations were generated from a seven-point, five parameter logistic standard curve, except for IL-13. Is there a particular reason for this or was it simply the assay standard?

6. A minor detail: the legend for Figure 2 labels d) as IL-12p70/IL-10 and e) as CCL2/IL-10, however these are reversed in the actual figure.

7. Since the authors set alpha at p<0.05, this is what should be considered statistically significant and there is no need for further statistical indices. In other words, p<0.004 is not “more significant” that p<0.05 as indicated in Figure 2.

8. Did any patients in either group require surgery or any other major invasive procedure? This would obviously skew the data on inflammation if so. Please address.

9. Were there any other outcomes assessed, such as 30d mortality, ventilator days, ICU length of stay, or hospital length of stay? This would make the data more robust.

Reviewer #3: PONE-D-19-33625

In this study, the investigators utilized multiplex cytokine kits to evaluate the cytokine signature of premature infants with late onset sepsis. The manuscript is well written without egregious spelling or syntax errors. I have some comments for the authors.

1. This study is not hypothesis driven and the cohort is extremely small making it difficult to draw any concrete conclusions. How will the results from the current study assist the providers that treat preterm neonates with sepsis?

2. The “no LOS” group was not clearly defined. Were these infants that displayed the signs and symptoms of sepsis ie increased respiratory support, temp instability, feeding intolerance or A's and B's but just did not have positive blood cultures? So by definition of the LOS group having positive blood culture and CRP>15 and >5 days of antibiotic therapy did that imply that you could have one or two of these criteria but not all three and be included in the “no LOS” group? Also, are the criteria utilized to define LOS utilized in the majority of neonatal literature examining sepsis?

3. Under the definition of LOS, I don't understand why four positive blood cultures were classified as no LOS (line 98). Please explain the rationale for this statement and criteria.

4. In the methods section for the description of normal values for the cytokine multiplex, please provide those data in a table (line 117-122).

5. How do the authors explain/justify that the IL-6/IL-10 ratio was not different between the two groups? How did they choose the ratios to examine; isn't there a nearly infinite number of combinations? In holding with this question, why did they choose IL-10 and not another anti-inflammatory cytokine for the examination described in Fig 2 (lines 198-204)?

6. In the first line of the discussion the authors use the term "very preterm". This term should be defined and is this term the nomenclature routinely utilized in the neonatal literature?

7. Since the authors did not see a difference in IL-6/IL-10 ratio and it is reportedly related to DIC in sepsis, they should investigate whether any of their LOS patients had DIC.

8. If the authors would have utilized the data from this study to try to predict which preterm infants at risk for LOS in a separate cohort, it would make this manuscript much more relevant.

6. PLOS authors have the option to publish the peer review history of their article (what does this mean?). If published, this will include your full peer review and any attached files.

Reviewer #1: No

Reviewer #2: No

Reviewer #3: No

---

## [Author Response · Author response to Decision Letter 0]

27 Mar 2020

Answer: Formatting of headings and sub-headings have been corrected.

2. In your Methods section, please provide additional information about the participant recruitment method and the demographic details of your participants. Please ensure you have provided sufficient details to replicate the analyses such as: a) the recruitment date range (month and year), b) a description of any inclusion/exclusion criteria that were applied to participant recruitment, c) a description of how participants were recruited, and d) descriptions of where participants were recruited and where the research took place.

Answer: Additional information about the participant recruitment method have been updated in the "Methods" section on page 4/5.

Reviewer #1: Title: Plasma cytokine profiles in very preterm infants with late-onset sepsis. Summary: Thank you for the opportunity to review this manuscript by Hibbert et al. The authors present an interesting study in which blood samples were obtained from 31 preterm neonates at the time of evaluation for late-onset sepsis. Plasma cytokines concentrations were determined, and inferences were made about the status of neonatal immune responses. The authors should be applauded for performing the study given the difficulties in blood collection in this vulnerable population, and the need for improved understanding of the neonatal immune response. Overall the study is very well presented, but could benefit from some clarification and potentially a multivariate logistic regression.

Major Points:

1. Why were patients with necrotizing enterocolitis excluded from the study? This is an important source of gram-negative, late-onset sepsis. Is this disease process considered fundamentally different from the LOS cases studied?

Answer: We agree that necrotising enterocolitis (NEC) is a very important potential source of infection and inflammation that requires further understanding, however, the pathogenesis of NEC and the relationship between NEC and LOS are unclear (reviewed in PMID: 25171544). Further, blood culture-positive NEC is most often associated with Gram-negative bacteria, for which infections are more severe and elicit a stronger inflammatory response (PMID: 22633519). There were two infants in our cohort who developed blood culture-positive NEC, one prior to the episode of LOS and one 12 days following a ‘no LOS’ episode. It is not within the scope (or power) of this study to evaluate the relationship or impact NEC has on subsequent LOS episodes, therefore the infant with the blood culture-positive NEC episode prior to the suspected LOS episode was excluded from our analysis. Exclusion of this infant from analysis has been clarified on page 4 of the revised manuscript.

2. In Methods, Blood sampling: blood was collected close to the time of blood culture sampling. How was blood collected and how much was drawn? Was the blood draw performed separately from the blood culture collection? Was it a capillary heelstick, venipuncture, or drawn from an existing line?

Answer: Thank you for bringing this to our attention. We collected 0.5ml of peripheral blood by venepuncture, with the majority of samples collected either at the time of sampling for blood culture (n=21) or 2-3 hours prior to sampling for culture (mean � SD, 124 � 177 min). The process for blood collection has been updated on page 5 of the revised manuscript.

3. Could the composite cytokine score ratio of anti- to pro-inflammatory cytokines be biased by the fact that there are only two anti-inflammatory cytokines (IL-13 and IL-10) included compared to many more (10) pro-inflammatory cytokines in the ratio? Since sums of the normalized scores are used for the ratio, it seems that the ratio will always be small given the large sums in the denominator for 10 different inflammatory cytokines.

Answer: We agree with the reviewer and have removed the composite cytokines scores and ratios from the manuscript.

4. Did the authors consider performing multivariate logistic regression to predict sepsis vs. non-sepsis? This is a well-performed prospective cohort study with fairly well-balanced sepsis vs. non-sepsis groups. It would be interesting to evaluate which clinical and cytokine factors emerge as the most significant predictors of sepsis, as this could help determine which cytokines should be measured to assist with the diagnosis of neonatal sepsis. If possible, I would suggest having sepsis vs. non-sepsis be the outcome of the regression, and include multiple explanatory variables (such as IL-10, IFN-gamma, gestational age at birth, day of life, CRP, platelet count). This would aid in creating a predictive model with an area-under-the-curve to predict late-onset sepsis based on parameters that can be obtained from POC tests/clinical information. This would also lend support to statements in the conclusion about the association of cytokine levels with sepsis vs. non-sepsis.

Answer: We agree with performing multivariate logistic regression in principle, however, following consultation with our biostatistician colleague, we were advised that this would require a larger sample size. With the current sample size, we believe that regression models would be unstable and findings would not necessarily be robust, and therefore, is beyond the scope of this project.

Minor Points:

1. Table 1 has a line describing “mortality during episode”. What does this mean, i.e. how is the episode defined? I would consider deleting this line given that there is already a descriptive statement in the results stating that only one infant died 12 days after the initial sample collection.

Answer: Thank you, we agree with the reviewer and have removed this variable from Table 1.

2. Table 2: Please include the calculated p-value rather than “ns” as this offers a more complete picture of the analysis. For example, median IL-15 levels were 43.8 in the sepsis group and 218 in the non-sepsis group. A p-value rather than “ns” here would help with interpretation of that result.

Answer: Thank you, the P values in Table 2 have been updated.

Reviewer #2: The manuscript, “Plasma cytokine profiles in very preterm infants with late-onset sepsis” evaluates various cytokine profiles in the plasma of very preterm infants. They found that there are marked innate inflammatory responses, with a relatively high anti-inflammatory to pro-inflammatory ratio. While the understanding of immune responses in preterm infants is important, a number of issues need to be addressed prior to publication:

1. Can the authors state very clear inclusion and exclusion criteria in the methods section? There is a mention in the limitations that the “no LOS” group were patients that had concerns for sepsis, but ultimately did not have a positive blood culture. Without further description, this is not necessarily an adequate comparison group. Was there any possibility to collecting blood from healthy pre-term infants or did the authors feel this was not clinically ethical? In addition, the authors state that one infant from the “no LOS” group had NEC. Did any of the other “no LOS” patients have an infection? This would be extremely important to know, as not all septic patients will have positive blood cultures and/or bacteremia may be transient enough to miss.

Answer: Thank you for raising these points. This was a prospective observational study of early life innate immune system ontogeny and all infants born less than 30 weeks’ gestational age without birth defects or genetic abnormalities and admitted to the NICU at King Edward Memorial Hospital were eligible for enrolment (updated on page 4 of the revised manuscript). The scope of this project, characterisation of cytokines at the time of sepsis, was not the primary aim of the main study, therefore collection of blood samples at the time of LOS evaluation were collected opportunistically. We could not ethically justify the collection of blood from infants who did not have a septic workup. The majority of very preterm have a multitude of co-morbidities, including chronic lung disease, intraventricular haemorrhage, retinopathy of prematurity, making it challenging to identify a cohort of infants that are ‘healthy’ for comparison. The ‘no-LOS’ group of infants represent the closest matched group to our sepsis group at the time of suspected sepsis and differ only for characterisation in the presence of detectable infection at that time. However, we can also confirm that no other infants, aside from the one infant who developed NEC 12 days following a ‘no LOS’ episode, in the no LOS group developed an infection during their admission to the NICU, this has been updated on page 7 of the revised manuscript.

2. Can the authors state why they chose a single early time point and what was the goal for sample collection? In the methods section, they state that, “hematological parameters were recorded close to the time of blood culture sampling” of about 3h (124 ±177min). Was the time of blood culture sampling presumably time zero of diagnosis of LOS?

Answer: In our study, and the majority of other neonatal sepsis studies, blood sample collection at the time of blood culture is common practice and is considered the onset of infection. For ethical reasons one blood sample at the time of blood culture was permitted, and has been clarified on page 5 of the revised manuscript. Abnormal WBC, neutrophil and platelet counts have been shown to be associated with the onset of sepsis and were therefore reported at the time of onset of LOS.

3. Although the authors state that 83% had gram positive cultures, can the remainder of the organism types be listed (number and %). Were there any causes for the bacteremia (ie what was the infectious source? Was this primary bacteremia or from other sources)?

Answer: Thank you, we have clarified the number and percent of the other causative organisms in Table 2. The infants in our cohort were born <30 weeks gestational age with a median birthweight of <1000g. Very preterm birth and very low birthweight are the leading factors contributing to the increased risk of LOS, while the infectious source is often unknown for neonatal late-onset sepsis (PMID: 20876594).

4. While the LOS group had significantly higher levels of both pro- and anti-inflammatory cytokines, this does not necessarily predict immune cell function and therefore one cannot conclude that the patients are “immunosuppressed”. Do the authors have any insight into actual immune cell function?

Answer: We agree with the reviewer, and as part of the original study characterising innate immune system ontogeny in preterm infants, we have found that the infants with LOS have impaired cytokine responses when challenged with live bacteria under ex vivo conditions. The manuscript describing these results has recently been published with Clinical Infectious Diseases (PMID: 31960030) and reference has been made in this manuscript on page 12 of the revised manuscript.

5. In the “multiplex immunoassay” section in the Methods, the authors state that cytokine concentrations were generated from a seven-point, five parameter logistic standard curve, except for IL-13. Is there a particular reason for this or was it simply the assay standard?

Answer: Five-parameter (5PL) and four-parameter (4PL) logistic modelling are the most widely accepted curve fitting methods for immunoassays (PMID: 24918306; PMID: 29536273). For S-shaped curves 4PL fits symmetrical data whereas 5PL has an added parameter that allows a better fit for asymmetrical data. Our IL-13 data was symmetrical and the 4PL model was used. The remaining cytokines/chemokines data were asymmetrical and fit using the 5PL model.

6. A minor detail: the legend for Figure 2 labels d) as IL-12p70/IL-10 and e) as CCL2/IL-10, however these are reversed in the actual figure.

Answer: Thank you for pointing this out, the figure legend has been corrected.

7. Since the authors set alpha at p<0.05, this is what should be considered statistically significant and there is no need for further statistical indices. In other words, p<0.004 is not “more significant” that p<0.05 as indicated in Figure 2.

Answer: Thank you, we understand the concern raised by the reviewer, but think it is important to provide the reader with an indication of the P value, as such we have updated the legend with the exact P value.

8. Did any patients in either group require surgery or any other major invasive procedure? This would obviously skew the data on inflammation if so. Please address.

Answer: We can confirm that during the first 4 weeks of life no infants in the cohort required surgery. Due to the prematurity of infants born <30 weeks gestational age a large proportion of the infants will experience some level of invasive intervention (e.g. mechanical ventilation, blood infusion) or medical condition (e.g. intraventricular haemorrhage) that may have an impact on inflammation, however, given the sample size of this study, only adjustment for a limited number of variables is feasible.

9. Were there any other outcomes assessed, such as 30d mortality, ventilator days, ICU length of stay, or hospital length of stay? This would make the data more robust.

Answer: We agree with your suggestion and have added several outcomes to Table 1, including duration of mechanical ventilation, duration of CPAP, intraventricular haemorrhage, ROP and length of NICU stay. 

Reviewer #3: In this study, the investigators utilized multiplex cytokine kits to evaluate the cytokine signature of premature infants with late onset sepsis. The manuscript is well written without egregious spelling or syntax errors. I have some comments for the authors.

1. This study is not hypothesis driven and the cohort is extremely small making it difficult to draw any concrete conclusions. How will the results from the current study assist the providers that treat preterm neonates with sepsis?

Answer: We acknowledge that the sample size is modest, reflecting the realities of a real-world study in this vulnerable population. However, but are confident that the findings from this study, especially the IL-10 and CCL2, results will prompt further investigation using larger studies to examine the utility of these cytokines as adjunct sepsis diagnostic makers to blood culture. The results from this study also add to and support the existing literature in this field, despite the acknowledged limitations.

2. The “no LOS” group was not clearly defined. Were these infants that displayed the signs and symptoms of sepsis ie increased respiratory support, temp instability, feeding intolerance or A's and B's but just did not have positive blood cultures? So by definition of the LOS group having positive blood culture and CRP>15 and >5 days of antibiotic therapy did that imply that you could have one or two of these criteria but not all three and be included in the “no LOS” group? Also, are the criteria utilized to define LOS utilized in the majority of neonatal literature examining sepsis?

Answer: Thank you, that is correct, a ‘no-LOS’ infant could have an elevated CRP, but without a positive blood culture. Currently, there is no consensus definition for neonatal sepsis (PMID: 26766602; PMID: 24751791), and positive blood culture is the accepted standard gold standard for diagnosing neonatal sepsis (PMID: 24751791). However, we also acknowledge that blood culture has limited sensitivity and specificity, the latter potentially leading to false-positive results (PMID: 24751791). For this reason, we included an additional objective marker, elevated CRP, as an indicator of inflammation and the clinical indicator of antibiotic therapy for �5 days as additional inclusion criteria, an approach previously used in analogous studies (PMID: 28367457, PMID: 16613997). Until there is a consensus definition for neonatal sepsis the classification will vary in the neonatal literature. Together with an international neonatal sepsis group we have highlighted the need for an improved and unified neonatal sepsis diagnosis (Pediatric Research, in press).

3. Under the definition of LOS, I don't understand why four positive blood cultures were classified as no LOS (line 98). Please explain the rationale for this statement and criteria.

Answer: Coagulase-negative Staphylococci (CoNS), a group of skin commensals, are the most commonly isolated group of organisms in very preterm infant late-onset sepsis. However, given these organisms are present on the skin, contamination of blood cultures is a frequent occurrence in the neonatal population leading to false-positive identifications of sepsis (3%-18%; PMID: 23331501, PMID: 11136522). These four blood cultures in our study were classified as contaminants based on the lack of an inflammatory response (no CRP rise within 48h of blood culture), negativity of a subsequent blood culture, and absence of clinical features of sepsis.

4. In the methods section for the description of normal values for the cytokine multiplex, please provide those data in a table (line 117-122).

Answer: Thank you for suggesting this, the multiplex minimum and maximum range has been added to Table 2 and removed from the methods section.

5. How do the authors explain/justify that the IL-6/IL-10 ratio was not different between the two groups? How did they choose the ratios to examine; isn't there a nearly infinite number of combinations? In holding with this question, why did they choose IL-10 and not another anti-inflammatory cytokine for the examination described in Fig 2 (lines 198-204)?

Answer: Elevated IL-6/IL-10 ratios have been shown in septic neonates with disseminated intravascular coagulation, which is commonly associated with Gram-negative infections. Our cohort of infants with Gram-positive LOS did not develop DIC and we feel this may contribute to the discrepancy in results – this has been clarified on page 13 of the revised manuscript.

We chose IL-10/TNF�, IL-6/IL-10 and IP-10/IL-10 ratios because they have been previously reported in adults and infants with infection as possible diagnostic markers, as referenced on page 10 of the revised manuscript (PMID: 10608764, PMID:27997530, PMID: 12719394). We observed a significant increase in CCL2, IFN� and IL-12p70 in infants with LOS and explored the ratio of these cytokines with IL-10. IL-13 was the only other anti-inflammatory cytokine in our panel, however the majority of levels were below the detection rate (common among neonatal sepsis PMID: 24013483; PMID: 29562764), therefore it was not a good candidate for exploring ratios. 

6. In the first line of the discussion the authors use the term "very preterm". This term should be defined and is this term the nomenclature routinely utilized in the neonatal literature?

Answer: Thank you, we have now defined this term on page 3 of the revised manuscript. The World Health Organisation sub-categorises preterm infants based on gestational age: extremely preterm (<28 weeks), very preterm (28-32 weeks) and moderate to late preterm (32-37 weeks) and this nomenclature is standard in the neonatal literature.

7. Since the authors did not see a difference in IL-6/IL-10 ratio and it is reportedly related to DIC in sepsis, they should investigate whether any of their LOS patients had DIC.

Answer: None of the infants with LOS in our cohort developed DIC, most likely related to the Gram-positive aetiology that predominated in our cohort. We have clarified this on page 13 of the revised manuscript.

8. If the authors would have utilized the data from this study to try to predict which preterm infants at risk for LOS in a separate cohort, it would make this manuscript much more relevant.

Answer: This is a valid point, and we agree in principle, however, this requires a larger sample size (as advised by a biostatistician colleague) and a separate independent validation cohort. This is beyond the scope of this project. We are currently conducting a larger neonatal study with a primary focus on sepsis, and anticipate completion in the next 2-3 years. We hope that other groups will also aim to validate our findings in other preterm populations.

---

## [Decision Letter · Decision Letter 1]

27 Apr 2020

Plasma cytokine profiles in very preterm infants with late-onset sepsis

PONE-D-19-33625R1

Dear Dr. Hibbert,

We are pleased to inform you that your manuscript has been judged scientifically suitable for publication and will be formally accepted for publication once it complies with all outstanding technical requirements.

With kind regards,

Philip Alexander Efron, MD, FACS, FCCM

Academic Editor

PLOS ONE

Additional Editor Comments (optional):

Reviewers' comments:

Reviewer's Responses to Questions

**Comments to the Author**

1. If the authors have adequately addressed your comments raised in a previous round of review and you feel that this manuscript is now acceptable for publication, you may indicate that here to bypass the “Comments to the Author” section, enter your conflict of interest statement in the “Confidential to Editor” section, and submit your "Accept" recommendation.

Reviewer #2: (No Response)

Reviewer #3: All comments have been addressed

2. Is the manuscript technically sound, and do the data support the conclusions?

Reviewer #2: Yes

Reviewer #3: Yes

3. Has the statistical analysis been performed appropriately and rigorously? 

Reviewer #2: Yes

Reviewer #3: Yes

4. Have the authors made all data underlying the findings in their manuscript fully available?

Reviewer #2: Yes

Reviewer #3: Yes

5. Is the manuscript presented in an intelligible fashion and written in standard English?

Reviewer #2: Yes

Reviewer #3: Yes

6. Review Comments to the Author

Reviewer #2: (No Response)

Reviewer #3: (No Response)

7. PLOS authors have the option to publish the peer review history of their article (what does this mean?). If published, this will include your full peer review and any attached files.

Reviewer #2: No

Reviewer #3: No

---

## [Editor Report · Acceptance letter]

4 May 2020

PONE-D-19-33625R1 

Plasma cytokine profiles in very preterm infants with late-onset sepsis 

Dear Dr. Hibbert:

I am pleased to inform you that your manuscript has been deemed suitable for publication in PLOS ONE. Congratulations! Your manuscript is now with our production department. 

With kind regards,

on behalf of

Dr. Philip Alexander Efron 

Academic Editor

PLOS ONE